# Towards Sustainable Human Resource Development of Convention Project Managers: Job Characteristics and Related Differences in Core Competency

**Yeonghye Yoon [1], Wenyan Yan [2] and Eunjin Kim [3,\*]**

[1] Global MICE Major, Dongduk Women's University, Seoul 02748, Korea; yhyoon@dongduk.ac.kr
[2] Korea Tourism Organization, Wonju 26464, Korea; ywy8897@knto.or.kr
[3] Department of Tourism & Convention, Pusan National University, Busan 46241, Korea
[\*] Correspondence: ejkim@pusan.ac.kr

**Abstract:** There are currently extensive discussions on the remarkable development of the convention industry and the impact on local economies made by convention visitors, but there is limited research on the human resources expertise which is crucial for sustainable and qualitative development in the field of convention. This study aims to examine differences in core competencies based on the type of job characteristics model of convention project managers. Based on the existing literature, quantitative and qualitative mixed-methods design was used. 12 semi-structured interviews were conducted with professionals to define measurements, especially core competencies. Job characteristics and core competencies were identified with a survey of 392 convention PM. By using a cluster analysis, it investigated the differences in perception of competencies according to the job characteristics model. As a result of the analysis, five job characteristics were verified, and core competencies that were not organized were classified. In addition, it was confirmed through the analysis that there is a difference in perception of core competencies according to job characteristics. Based on the results, this study stresses the importance of the qualitative and sustainable development of the convention industry. Theoretical and practical implications were provided to enhance core competencies according to job characteristics for sustainable growth of convention project managers.

**Keywords:** convention project manager; job characteristics model; core competency; cluster analysis

## 1. Introduction

The convention industry is recognized as a high value-added industry in that it promotes the image of a city and country holding a convention and creates added values influencing various industries related to the economy of the city or country [1]. Convention industry is a comprehensive term for MICE (meetings, incentive tours, conventions, and exhibitions), and is a rapidly growing part of the tourism field [2]. The convention industry exerts economic, political, and social influences on the country, and city tends to gradually expand throughout the world. For this, the South Korean government makes proactive efforts to develop and support the convention industry. As a result, South Korea was honorably ranked first in 2016, first in 2017, second in 2018, and third in 2019 in the UIA's Global Meetings ranking [3]. In the situation of South Korea, where there are insufficient resources and the knowledge industry is being developed, the convention industry is one of the fastest growing industries and has become an important source of future growth.

However, the convention industry in South Korea has developed at a much faster pace than any other country, so there is a lot of skepticism on its qualitative growth. Despite the increases in convention infrastructure and hardware, the number of holding international conferences, and the

number of overseas visitors, the qualitative growth, which means competitiveness of industry, is lagging behind with poor content, lack of professional manpower, and ecosystem topics [4]. The convention industry is especially recognized as an area where machines could not replace humans. The task of holding an international conference is relatively unstructured work including meeting the host's needs, and in this regard, machines could not replace human workers. Although the importance of human resources is emphasized, academic research has yet to address the issues of the job characteristics and competencies [5,6]. Most studies of human resources and jobs in the convention field are on job satisfaction and turnover intention, as well as loyalty in the organization [7–11].

Meanwhile, work of in the convention industry is carried out on a project basis. There are different time periods for planning, operating, and finishing convention events; however, each shows a business pattern as a project. In a project, the business is conducted in the following process: "Beginning-Planning-Execution-Monitoring & Control-Completion", and the role of a project manager is crucial. The project manager is required to have a variety of knowledge concerning the organization's business objectives and initiatives, takes a role for monitoring rather than execution, and has limited authority for decision-making [12]. Along with this, since project management always performs a new project, it is difficult for them to get accustomed, and so there is uncertainty. Various persons are concerned, and the clients' requirements change according to times and situations. Accordingly, the role of the project manager who leads a project cannot be but very difficult. And yet, an enterprise should decide which project produces performance for the organization and is worth investing in; once decided, since it is necessary to manage the project strictly by preparing a clear system, the role of the project manager becomes more important [13]. Of 10 knowledge areas concerning what to manage in the project, human resources management plays a very important role [14]. Especially in the convention industry, in which the roles and achievements of human resources are very important, for sustainable development and growth, studies of human resources and jobs should be continuously conducted.

Recognizing the necessity of research on convention human resources, this study delves into the issues of human resources management and jobs. Of various theories on human resources management, this study would conduct research based on "the Job Characteristics Model (JCM)" of Hackman & Oldham [15]. The job characteristics theory has been widely utilized in various human resources management studies in social sciences, including tourism and hotel, but it has hardly been integrated into the convention field yet. The job characteristics theory has been developed, drawing attention as a research topic since 1965, and in the early stage, job attributes were classified into variety, autonomy, operational interaction, knowledge and skill, and responsibility. These job attributes affect employees' motivation and job satisfaction [16].

Consequently, as the theory of job characteristics leads to job satisfaction and performance, the necessity of research is raised in the dimension of human resources management in the organization [17, 18]. The difference of this study is that it is an initial attempt to introduce the theory of job characteristics for convention project managers in the convention field where research on human resources is insufficient. In addition, in the field of human resource management and development, it is possible to derive academic implications that can be strengthened by verifying theories. Thus, through this study, checking project managers' perceptions of job characteristics in the convention field and discovering differences in core competencies according to each type of job characteristics could contribute to the increase of performance and effectiveness in the sustainable organization. Research and effort for human resources development can secure competitiveness and sustainable growth in the convention field, as well as a logic responding to the limitations, and where quantitative growth and the voice of criticism can be heard.

## 2. Theoretical Background

### 2.1. Job Characteristics of Convention the Project Manager

This study aims to confirm the convention project manager's job characteristics based on the characteristics of the convention business, referring to five job characteristics of Hackman & Oldham [15] presented in Figure 1.

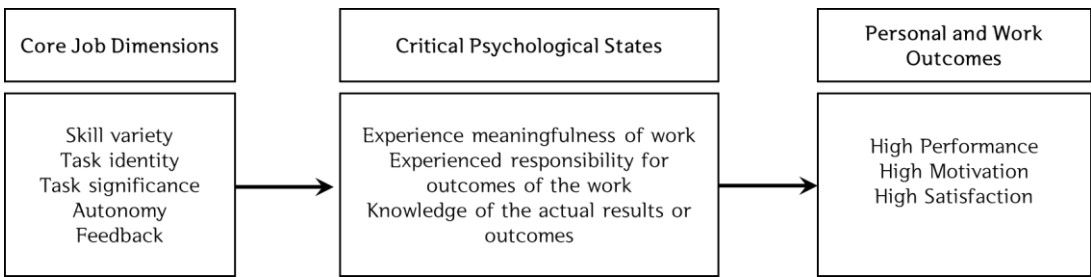

**Figure 1.** Job Characteristics Model.

Job characteristics have been developed, receiving attention as a research topic since 1965. In the early stage, job attributes were classified into variety, autonomy, interaction, knowledge and skill, and responsibility, and these job attributes affect the employees' motivation and job satisfaction [16]. Based on this study, Hackman and Lawler [19] developed a job characteristics measurement tool called Yale job inventory (YJI) to measure the job characteristics affecting the employees' behavioral responses. YJI became the base of Job diagnostic survey (JDS) developed by Hackman and Oldham [20] later, and then, Hackman and Oldham developed the JCM which is the dominant model of work design. Job characteristics refer to the dimensions of the characteristics of core duties that provide perceptions about the significance, responsibility, and result felt while individually performing the job [21]. Research on job characteristics is also utilized as an index for job analysis. Since the duties differ depending on the organization's characteristics, types, and businesses, and the performance may differ depending on the characteristics of the job performed, the HR personnel and managers should make efforts to understand job characteristics properly. Accordingly, research to measure job characteristics is continuously conducted, defining them more appropriately [22,23]. In the JCM, job characteristics consist of five dimensions, including skill variety, task identity, task significance, autonomy, and feedback [17,24]. Skill variety, task identity, and task significance make the employees feel the significance of the job; autonomy makes them experience a sense of responsibility for the job; and feedback makes them experience a psychological state to perceive the actual result of business activities. In other words, the JCM posits that job characteristics affect job satisfaction and performance, motivating job performance by strengthening the psychological state [25–28].

Despite that the JCM is still in full use today with abundant preceding studies, there are criticisms on the research [22]. Some scholars pointed out a problem in the factor structure and low internal consistency of variables [29,30]. Also, it is suggested that the characteristics dimension should be looked from more comprehensive perspectives to cover many other tasks because of the emergence of new jobs and changes of work environments [22,31]. Morgeson & Humphrey [22] developed the Work Design Questionnaire (WDQ) to measure job characteristics more comprehensively in modern work environments. It assessed work characteristics included in the existing model of JCM (work and knowledge domains), and social and contextual characteristics examined in the previous studies. However, despite the contemporary perspective of work design, the JCM is still a highly dominant and influential model in work design [23].

A professional conference organizer is one who attracts, plans, and operates convention events, providing the professional convention business to organizers such as the government, associations, enterprises, and non-profit organizations, through contracts for a certain period of time on the project base. A project is an activity performed to create a profitable result for a limited period of time with

a goal. PCO performs the convention project manager's role, since they attract, plan, and perform events in a limited time and lead to successful results. Recently the convention project managers' work area tends to expend to other service providers including convention centers, event and promotional companies, travel agencies, convention bureaus, and private companies, for which they are directly attracting and planning events. As function of the convention industry expands and becomes complex, the professionalism of HRM is required, but the research in job and competency is somewhat limited [32]. Certain empirical studies in the hospitality and convention area examine the relation between job characteristics and work results [28]. Considering qualitative growth of the convention project manager, it is crucial to observe their job characteristics and competency through characteristics.

As mentioned above, most studies of job characteristics have been conducted, focusing on the relationship with positive job attitude based on the JCM of Hackman and Oldham [21]. The job characteristics model was created to be applicable to the business performed somewhat independently [20]. The design is considered to be appropriate for understanding the job characteristics of the convention project manager who independently performs business on a project basis. To summarize the above contents, the convention project manager's job characteristics can provide motivation and satisfaction through the convention business, and they can be presented with the attributes of the job perceived by the project manager. Therefore, this study would examine the convention project manager's job characteristics leading to organizational performance and analyze impacts on the dependent variables according to the types of perceptions of them.

## 2.2. Competencies of Convention Project Manager

In a rapidly changing environment, the growth of the convention industry is remarkable, the importance and necessity of the promotion of the convention personnel in terms of professionalism has increased [33–35]. It is urgently necessary to develop human resources' qualitative growth and competency-based human resources management for sustainable growth of the industry from the pattern of development, making the organization's growth a priority in the past. In this sense, this study would investigate studies of the competency generally utilized, and examine the limitations of studies of convention competency.

Competency refers to the characteristic inherent in the person who achieved excellent and effective performance in the business [36,37]. McLagan [38] noted that competency has various meanings and can be divided into the areas related to the job, such as business, results, and product or areas related to the individual's characteristics such as knowledge, skill, and attitude. Spencer & Spencer [37] defined competency as a hidden characteristic in an individual to perform efficiently in various situations. Thus, the project manager's core competency is a compound of knowledge, skill, and attitude, that is needed for successful and efficient job performance [36,39,40]. Katz [41] classified competences for managers into three areas, including conceptual, human, and technical. Spencer & Spencer [37] divided competency into five, including motive causing behaviors; trait, which is a physical characteristic and situation; self-concept, showing an individual's attitude and value; knowledge, meaning an individual's information in a certain field; and skills showing the ability to perform a task. It is easy to develop skills and knowledge through education; however, it is hard to achieve or develop inherent competencies such as self-concept, motive, and trait through learning.

Meanwhile, much utilized in practices, Project Management Institute (PMI) in the U.S. developed a project manager competency development system. They present the project managers' competency in three areas; the knowledge area they should know concerning a project, the performance area concerning what they can get applying the project management knowledge, and the individual area concerning how they should act in performing a project. And yet, the scope of project managers is various and competencies are similar in a large scheme, but it needs to be more specified when applied to each industry. It shows that professionalism is required beyond administrative and knowledge professionalism suggested by PMI [39,42]. For example, with the competency model of Boyatizs [36] and Spencer & Spencer [37], Crawford [39] suggests considering knowledge, skills, demonstrable

performance, core personality characteristics, and personal characteristics as competencies. Morris [43] emphasizes that project managers' competency should be beyond administrative professionalism of setting and managing the scope, timeline, and budget, and posits that they should have problem-solving professionalism, leadership, content knowledge, analytical skills, and communication skills.

Brill, Bishop, & Walker [42] criticized that only the parts of knowledge and performance were mostly emphasized in the criteria for project competencies provided by project management associations of the major countries, including Project Management Body Of Knowledge (PMBOK) in the U.S., the Association for Project Management (APM) in the U.K., and the Australian Institute of Project Management (AIPM) in Australia. They also emphasized that project managers' competencies should be figured out through an empirical study. In their study, leadership professionalism and problem-solving appear to be the most important categories of competency, followed by project administrative professionalism, content knowledge, communication professionalism, personal professionalism, and analytic professionalism.

In the literature of core competencies in hospitality, tourism, and convention, some are conducted with the convention industry, but most surveys are conducted with the tourism and hospitality field [33]. However, since the convention industry has complicated task processes in which more diverse persons and service providers participate than the hospitality industry does, differentiated competencies such as knowledge, skills, and ability (KSA) [44] are required. In the preceding studies, there are fewer studies with the convention industry, but some studies are conducted to draw the competency suitable for the situations and realities of the convention industry. However, most are competencies to perform general tasks, which are insufficient for understanding core competencies in terms of project management.

In cooperation with the Canadian Tourism Human Resources Council (CTHRC), Member Professional International (MPI) applied Event Management International Competency Standards (EMICS) developed by them to develop standard competencies for business events, and professionals from 20 countries participated in developing Meeting and Business Event Competency Standards (MBECS) in 2011 [45]. This includes 12 major competencies to perform business event tasks, including strategic planning, project management, risk management, financial management, administration, human resources, stakeholder management, meeting or event design, site management, marketing, professionalism, and communications. For a model of international convention professionals' competencies, Tang [34] conducted the Delphi technique with 11 professionals, and as a result, divided competencies broadly into communication competence, intellectual competence, and professional activity competence. And yet, the research conducted with English education, tourism, and international business professionals, there is a limitation in the composition of the panel. Thus, it is necessary to draw core competencies more multilaterally and more in depth with convention professionals and convention employees in understanding project managers' core competencies.

The biggest concern for creating competitive advantage and drawing positive performance in the industry is, what is the most important competency for convention project managers [42]? In the convention business, it is crucial to have various competencies such as problem-solving professionalism, leadership, content knowledge, analytic skills, and communication skills beyond administrative professionalism for the convention project managers who run the entire event. Thus, investigating the differences in individual core competencies according to the types of convention project managers' job characteristics would present important directions of education and continuous learning for improving competencies in the future.

## 3. Materials and Methods

### 3.1. Research Questions

The purpose of this study was to define the key variables for the job characteristics and core competencies of convention project managers, and to identify core competencies based on the type of

job characteristics model of convention project managers. In order to do this, the study was designed to answer the following research questions:

Q1. Which factors are identified as the job characteristics and core competencies of convention project manager?

Q2. Are the core competences of convention project managers different among groups of job characteristic?

Q3. Are the core competencies of convention project managers different among groups of job characteristics?

Q4. Are the demographics of convention project managers different among groups of job characteristics?

### 3.2. Generation of Questionnaire Items

Items reflective of job characteristics and core competencies were developed through an iterative process involving three steps of item generation. In the first step, items from the literature were reviewed, including job characteristics [20–22] and core competencies [39,42] of the project manager in general. Based on the review of literature, a total the existing items were reviewed and revised for application within the convention project manager.

The second step was the crucial part of the process to gather the insights generated from conducting interviews with experts in the convention industry. A series of interviews were conducted with professional convention personnel and convention center managers who had at least 10 years of experience as convention project managers. The selection of participants was conducted using the snowball method and each expert was asked to recommend potential participants for the interviews. The interviews were semi-structured and the data were transcribed and analyzed to modify preliminary items, as well as to determine new applicable items which explain job characteristics and core competencies well. A total of two interviews were conducted based on previous studies. In the first interview, a total of 20 convention project managers were confronted to collect opinions on core competencies. After coding and analyzing the data obtained through this, the core competencies were confirmed through 12 people in the second interview.

The last step in the sample item development process was to purify the measurement instruments. A pretest was conducted with 100 practitioners such as project managers in the convention industry including convention center, destination marketing organization (DMO), tour company, convention service providers, etc. They were members of Korea MICE Association, as well as members of Korea MICE Alliance in Korea tourism organization. The pretest data were used to evaluate the construct validity and reliability of the measurement items. Upon completion of pretest data analysis, the measurement items were then finalized and compiled into survey instruments to utilize for data collection. Finally, the questionnaire of this study was composed by extracting 52 questions centering on variables derived from previous studies. The questions related to job characteristics consisted of 22 questions, the core competency consisted of 18 questions, the organizational effectiveness consisted of a total of 12 questions, and the questionnaire consisted of 52 questions. Each item was analyzed using the likert 5-point scale, and six items were constructed using a nominal scale and a ratio scale to understand the demographic characteristics of respondents. The questionnaire items on job characteristics brought five items (skill variety, task identity, task importance, autonomy, feedback) from Hackman and Morgeson's study. The core competency questionnaire was composed of four items (crisis management, event attraction, task management, and communications) that were a result of previous studies and two interviews.

### 3.3. Data Collection and Analysis

The main survey was distributed by both onsite and online survey. The survey was conducted from 1 December 2015 to 20 February 2016. The samples were convention project managers in the field of convention. The onsite survey was conducted throughout Korea MICE Alliance convention held in

December 2015. On the day of the convention, the survey was conducted through self-administered questionnaires, and the data were collected with trained researchers. For the online survey, the invitation to participate in the survey was mailed to practitioners mentioned above. It was conducted to alleviate the problems of location and time constraints. They were convention project managers in the field of convention drawn from a membership directory of the Korea MICE Association and Korea MICE Alliance in Korea tourism organization.

From a total of 600 surveyed, 392 responses were collected. As a result of data screening, valid responses were included in the final data analysis, yielding a total response rate of 65.3%. In order to analyze the collected data, Statistical Package for the Social Sciences (SPSS) 21 version was used. Prior to the data analysis, data sets screening was performed to verify the violations of assumptions. T-test was performed to verify the differences in onsite and online survey results. After confirming that the difference between the means of the two groups was not significant in almost all variables, a total of 392 valid samples were used for the final analysis. The collected data were analyzed by conducting frequency analysis to understand the characteristics of the data. To evaluate the validity and reliability, principal factor analysis with varimax rotation was conducted on all statements. The hierarchical cluster analysis using the Ward method and the K-means cluster analysis were combined to subdivide the group according to the job characteristics of the convention project manager. Discriminant analysis was then applied to detect whether any significant differences existed among the three groups. Cross-tabulations analysis was performed to verify the difference in demographic profile according to the job characteristics of the convention project manager. Finally, multivariate analysis of variance (MANOVA) was conducted to verify the difference in core competencies for each group according to the recognition of job characteristics of convention project managers.

## 4. Results

### 4.1. Demographic Distribution

Demographic characteristics of respondents were presented in Table 1. Of the participants, 59.4% were female, whereas 41.6% were male. Of the 392 respondents, 44.9% were in their 30s, 23.7% were in their 20s, 23.0% were in their 40s, and the rest were in their 50s. The largest category for monthly income (38.5%) was between above 1700–below 2550 US$, followed by below 1700 US$ (20.2%), above 2550–below 3400 US$ (20.2%), and above 4200 US$ (12.5%). For their affiliation, the large number of respondents were Convention planning company (50.5%), followed by convention center (19.4%), convention service provider (9.4%), convention bureau (8.2%), and travel company (5.4%). The majority of the participants (94.6%) held at least a bachelor's degree, while the rest had completed two-year college (5.4%).

**Table 1.** Demographic characteristics.

| | Item | Frequency | Ratio | | Item | Frequency | Ratio |
|---|---|---|---|---|---|---|---|
| Sex | Female | 229 | 58.4 | | Convention planning company | 198 | 50.5 |
| | male | 163 | 41.6 | | Convention center | 76 | 19.4 |
| Age | 20–29 | 93 | 23.7 | | Convention Bureau | 32 | 8.2 |
| | 30–39 | 174 | 44.8 | Affiliation | Travel agency (convention team) | 21 | 5.4 |
| | 40–49 | 90 | 23.0 | | Convention service company | 37 | 9.4 |
| | 50–59 | 31 | 8.0 | | | | |
| Monthly income (USD) | below 1700 | 79 | 20.2 | | Others | 28 | 7.1 |
| | above 1700–below 2550 | 151 | 38.5 | | | | |
| | above 2550–below 3400 | 79 | 20.2 | | College | 21 | 5.4 |
| | above 3400–below 4200 | 34 | 8.7 | Education | University | 236 | 60.2 |
| | above 4200 | 49 | 12.5 | | Graduate school | 135 | 34.4 |
| Marital Status | Yes | 179 | 45.7 | | | | |
| | No | 213 | 54.3 | | | | |

## 4.2. Cluster Analysis Based on the Job Characteristics of the Convention PM

In this study, exploratory factor analysis was conducted to derive the dimension of the concept and test reliability of the items. As a result of principal component analysis (PCA) with varimax rotation methods, eigenvalues of 1 or more, and factor loadings of 0.4 or more were derived as factors. The Kaiser-Meyer-Olkin (KMO) measure and Bartlett's test of sphericity were conducted to determine the appropriateness of factor analysis.

Table 2 represents the results of factor analysis of the convention project managers' job characteristics. First, exploratory factor analysis for job characteristics was performed using the PCA with varimax rotation methods. Two items were removed due to low primary factor loading (less than 0.4) or irrelevant items. The remaining 22 items loaded on five factors: *professionalism/skill variety, autonomy, task identity, task importance, and feedback*. The results of KMO measure of sampling adequacy appears a value of 0.900 and Bartlett's test sphericity demonstrates a significance at a level of 0.000 ($\chi^2$ = 4868.243, df = 231).

**Table 2.** Data reliability and feasibility analysis.

| | Factor | Factor Loading Value | Cronbach's $\alpha$ | Eigen Value | Variance Explained |
|---|---|---|---|---|---|
| professionalism/ skill variety | Specialized knowledge and skills | 0.865 | 0.914 | 4.227 | 19.214 |
| | Depth of knowledge and expertise | 0.850 | | | |
| | A high degree of professionalism in terms of purpose and activity | 0.833 | | | |
| | Expertise in tools and procedures | 0.779 | | | |
| | Advanced technology and knowledge | 0.735 | | | |
| | Advanced technology and knowledge | 0.676 | | | |
| autonomy | Proactively decide how to do business | 0.836 | 0.878 | 2.982 | 13.553 |
| | Freely decidable tasks | 0.825 | | | |
| | Have discretion to work | 0.819 | | | |
| | can decide work independently | 0.691 | | | |
| task identity | mostly done from start to finish | 0.792 | 0.842 | 2.768 | 12.580 |
| | Responsible for the entire process of the task | 0.780 | | | |
| | Carry out the entire task | 0.763 | | | |
| | Oversee and manage the entire project | 0.697 | | | |
| task importance | In the absence of this work, disadvantages | 0.818 | 0.821 | 2.666 | 12.119 |
| | Loss in case of business processing error | 0.748 | | | |
| | This task is important to the company | 0.740 | | | |
| | The result of business processing has a great influence on colleagues | 0.654 | | | |
| feedback | Possible to recognize the error part during task | 0.765 | 0.785 | 2.501 | 11.369 |
| | Performance evaluation is possible at the end | 0.744 | | | |
| | Can recognize the current situation during business processing | 0.725 | | | |
| | Result evaluation is possible after the end of task | 0.708 | | | |

KMO = 0.900, Bartlett's sphericity test (Approximate chi-square = 4868.243, Degrees of free = 231, *p* < 0.001).

The first factor, named *professionalism/skill variety*, accounts for 19% of total variance and is associated with professional job knowledge, skills, techniques, etc. The second factor was named *autonomy* and explains 14% of total variance. This factor is related to having decision-making powers. The third factor, named *task identity*, is related to the degree to which the unit of work assigned to the convention project managers at the overall level, representing 13% of total variance. The fourth factor, named *task importance*, is associated with the degree to the importance of job in the company whole, representing 12% of total variance. The last factor, *feedback*, accounts for 11% of total variance and represents the degree to recognize the work situation and process in-between and after. All Cronbach's alpha coefficients are above 0.6, representing the scales reliable.

Table 3 signifies the results of factor analysis of job competencies. The results suggest that two items should be removed due to low primary factor loading (less than 0.4) or irrelevant items. The results of KMO measure of sampling adequacy appears a value of 0.895 and Bartlett's test sphericity demonstrates a significance at a level of 0.000 ($\chi^2$ = 2926.296, df = 153).

**Table 3.** Data reliability and feasibility analysis.

| | Factor | Factor Loading Value | Cronbach's $\alpha$ | Eigen Value | Variance Explained |
|---|---|---|---|---|---|
| crisis management | Ability to solve and cope with unexpected problems | 0.846 | 0.855 | 3.813 | 21.184 |
| | Recognize situations quickly in case of sudden problems and crises | 0.817 | | | |
| | When problems arise, solve them calmly without panic | 0.784 | | | |
| | Risk management team members and organizational management skills | 0.561 | | | |
| | Decision making after thinking about alternatives | 0.506 | | | |
| event attraction | Sales and marketing skills | 0.821 | 0.810 | 3.045 | 16.917 |
| | Negotiation ability | 0.748 | | | |
| | Presentation skills | 0.677 | | | |
| | Identifying and applying social trends | 0.673 | | | |
| | Willingness to develop common sense skills | 0.624 | | | |
| task management | Ability to accurately calculate work scope | 0.722 | 0.772 | 2.419 | 13.441 |
| | Convention expertise | 0.604 | | | |
| | Budget planning and management skills | 0.600 | | | |
| | Schedule coordination and adjustment ability | 0.567 | | | |
| | Program and content development ability | 0.542 | | | |
| communications | Approach in the way preferred by the relevant stakeholders when communicating | 0.661 | 0.641 | 1.688 | 9.375 |
| | Actively reflect the opinions of project team members when making decisions | 0.652 | | | |
| | Solve face-to-face when actively listening and communicating with stakeholders | 0.577 | | | |

KMO = 0.895, Bartlett's test of sphericity (Approximate chi-square = 2926.296, df = 153, $p < 0.001$).

Four factors have arisen with the remaining 18 items loaded on five factors: *crisis management, events attraction, task management, and communications.* The first factor, named *crisis management*, is concerned with ability to solve and cope with unexpected crisis or situations, representing 21% of total variance. The second factor was named *events attraction* and explains 17% of total variance, and related the ability needed to attract various convention events including marketing, sales, negotiation ability, etc. The third factor, named *business administration*, is associated with administrative ability as a whole, and explains 13% of total variance. The fourth factor, named *communications*, is related to communications skills with various stakeholders, representing 9% of total variance. The scales were considered as reliable because Cronbach's alpha coefficients are around 0.6 and more.

### 4.3. Cluster Analysis Based on the Job Characteristics of the Convention PM

Both the hierarchical and non-hierarchical cluster analysis methods were used for the cluster analysis based on the job characteristics of convention project manager. To determine the number of clusters, three fits solutions were obtained by referring to the increase in the similarity coefficients from step 386 to 388 in the agglomeration schedule. Then the average value of the hierarchical cluster analysis was set to initial seed and cluster analysis was conducted. This is because the number of clusters can be determined based on the similarity distance criteria and the average value of input variables for each cluster.

　　　MANOVA was performed between the job characteristics of the convention project manager and the clusters. The result represents in Table 4: Cluster 1 (n = 143) is high level of job characteristic awareness group, Cluster 2 (n = 90) is low level of job characteristic awareness group, Cluster 3 (n = 156) is moderate level of job characteristic awareness group. To verify the cluster validity of the convention project manager's job characteristics, a discriminant analysis (Table 5) was performed to confirm that the clusters were correctly classified. All groups were found to be correctly allocated. In the case of Cluster 1, 95.1% of their cases were correctly allocated. Cluster 2 was found to be correctly allocated in 98.9% of their cases, and Cluster 3 was in 97.4%. Table 6 represents the results of MANOVA to confirm for any significant differences among clusters. The results of MANOVA show that the different groups had significant differences in their awareness toward the job characteristic.

**Table 4.** Cluster analysis result.

| | | Professionalism/ Skill Variety | Autonomy | Task Identity | Task Importance | Feedback |
|---|---|---|---|---|---|---|
| k-means cluster analysis | cluster 1 (N = 143) | 4.12 | 4.18 | 4.60 | 4.43 | 4.18 |
| | cluster 2 (N = 90) | 2.94 | 2.75 | 3.24 | 3.22 | 3.31 |
| | cluster 3 (N = 156) | 3.59 | 3.08 | 4.20 | 4.04 | 3.63 |
| | F | 122.954 *** | 210.256 *** | 179.350 *** | 158.247 *** | 80.756 *** |
| Sheffe | III | *** | *** | *** | *** | *** |
| | II–III | *** | *** | *** | *** | *** |
| | I–III | *** | *** | *** | *** | *** |
| Pillai's Trace = 0.949 *** Wilks' lambda = 0.183 *** | | | | Hotelling's Trace = 3.746 *** Roy's largest root = 3.542 *** | | |

*** $p < 0.001$, cluster1: High level perceptions, cluster2: Low level perceptions, cluster3: Moderate level perceptions.

**Table 5.** Discriminant analysis result.

| Function | Eigenvalue | Dispersion (%) | Canonical Correlation | Wilks Lamda | $\chi^2$ |
|---|---|---|---|---|---|
| 1 | 3.542 | 94.6 | 0.883 | 0.183 | 652.333 *** |
| 2 | 0.204 | 5.4 | 0.411 | 0.831 | 71.197 *** |
| cluster title | | cluster 1 (N = 143) | cluster 2 (N = 90) | cluster 3 (N = 156) | Total |
| High level perceptions | | 136 (95.1%) | 1 (0.7%) | 6 (4.2) | 143 |
| Low level perceptions | | 0 (0.0%) | 89 (98.9%) | 1 (1.1%) | 90 |
| Moderate level perceptions | | 3 (1.9%) | 1 (0.6%) | 152 (97.4%) | 156 |

*** $p < 0.001$, Hit ratio = 96.9%.

**Table 6.** Analysis result of differences between clusters by recognition of job characteristics.

| | Professionalism/ Skill Variety | Autonomy | Task Identity | Task Importance | Feedback |
|---|---|---|---|---|---|
| cluster 1 (N = 143) | 3.626 (0.511) | 3.262 (−0.643) | 4.186 (−0.524) | 4.008 (−0.524) | 3.669 (−0.522) |
| cluster 2 (N = 90) | 4.224 (−0.5) | 4.235 (−0.472) | 4.657 (−0.411) | 4.53 (−0.461) | 4.256 (−0.522) |
| cluster 3 (N = 156) | 2.89 (−0.621) | 2.64 (−0.632) | 3.315 (−0.762) | 3.272 (−0.762) | 3.298 (−0.522) |
| F-value | 156.884 *** | 191.891 *** | 146.884 *** | 153.899 *** | 88.422 *** |
| I–II | *** | *** | *** | *** | *** |
| II–III | *** | *** | *** | *** | *** |
| I–III | *** | *** | *** | *** | *** |

Pillai's Trace = 0.949 ***, Wilks' lambda = 0.183 ***, Hotelling's Trace = 3.746 ***, Roy's largest root = 3.542 ***
*** $p < 0.001$.

### 4.4. MANOVA of Segmentation Groups and Core Competency Factors

MANOVA was performed to investigate the differences among risk management, events attraction, business administration, and communications factors by the subdivision groups of the job characteristics of convention project manager (Table 7). As a result, Pillai's trace = 0.265 ($p < 0.001$), Wilks' lambda = 0.735 ($p < 0.001$), Hotelling's trace = 0.359 ($p < 0.001$), Roy's maximum root = 0.355 ($p < 0.001$). The core competencies factors were found to be significantly different among job characteristics clusters.

**Table 7.** MANOVA results between segmentation groups and core competency factors.

| | Crisis Management | Events Attraction | Task Management | Communications |
|---|---|---|---|---|
| cluster 1 | 4.402 | 4.009 | 4.104 | 4.131 |
| (N = 143) | (0.036) | (0.041) | (0.037) | (0.036) |
| cluster 2 | 4.628 | 4.288 | 4.386 | 4.379 |
| (N = 90) | (0.046) | (0.051) | (0.046) | (0.045) |
| cluster 3 | 4.140 | 3.686 | 3.656 | 3.810 |
| (N = 156) | (0.053) | (0.060) | (0.053) | (0.053) |
| F-value | 24.527 *** | 29.191 *** | 53.917 *** | 33.624 *** |
| I–II | *** | *** | *** | *** |
| II–III | *** | *** | *** | *** |
| I–III | *** | *** | *** | *** |
| Pillai's Trace = 0.265 ($p < 0.001$) Wilks' lambda = 0.735 ($p < 0.001$) | | Hotelling's Trace = 0.359 ($p < 0.001$) Roy's largest root = 0.355 ($p < 0.001$) | | |

*** $p < 0.001$, cluster 1: High level perceptions, cluster 2: Low level perceptions, cluster 3: Moderate level perceptions.

The group with lowest perceptions of job characteristic (Cluster 2) showed highest importance for the core competencies factors, followed by the group with highest perceptions (Cluster 1), and the moderate perceptions (Cluster 3) had the lowest importance.

### 4.5. Cross Tabulation of Job Characteristics Segmentation and Demographic Characteristics

Table 8 represents the result to verify the difference in demographic profile by the job characteristics of convention project manager. The cross-tabulation result showed statistical significance in gender ($\chi^2 = 13.607$, $p < 0.01$), marital status ($\chi^2 = 29.838$, $p < 0.001$), education ($\chi^2 = 21.263$, $p < 0.001$), monthly average income ($\chi^2 = 75.001$, $p < 0.001$), and age ($\chi^2 = 59.964$, $p < 0.001$).

Cluster 1 had more male (27.3%) than female (19.8%) respondents. The respondent rate of unmarried (11.1%) was higher than that of married (20.9%). In this group 27.8% of respondents were university graduates, 19.4% had average monthly income of above 1700–below 2550 US$, and the largest number of respondents were in their 30s. Cluster 2 had slightly more female (15.7%) than male (14.2%) respondents. In this group, there were more married people (18.8%) than that of unmarried (11.1%). The majority of respondents were university graduate (14.2%) or higher (14.2%), and they were in their 30s (12.4%) and 40s (11.1%). The average monthly income of this group was relatively higher than other groups with 7.3% of respondents who made more than 4200 US$. Cluster 3 had a similar demographic pattern as Cluster 1, except that the majority of respondents were 20s and 30s, which represents relatively low average monthly income among other groups.

**Table 8.** Job characteristics segmentation and demographic characteristics chi-square analysis result.

| Item | | Cluster 1 | Cluster 2 | Cluster 3 | Total | *p* |
|---|---|---|---|---|---|---|
| Sex | Male | 106 (27.3%) | 55 (14.2%) | 65 (16.8%) | 162 (41.8%) | $\chi^2$ = 13.607 |
| | Female | 77 (19.8%) | 61 (15.7%) | 24 (6.2%) | 226 (58.2%) | $p < 0.01$ ** |
| Marital Status | Yes | 102 (26.2%) | 43 (11.1%) | 67 (17.3%) | 212 (54.6%) | $\chi^2$ = 29.838 |
| | No | 81(20.9%) | 73 (18.8%) | 22 (5.7%) | 176 (45.4%) | $p < 0.001$ *** |
| Education | College | 10 (2.6%) | 6 (1.5%) | 5 (1.3%) | 21 (5.4%) | $\chi^2$ = 21.263 |
| | University | 108 (27.8%) | 55 (14.2%) | 69 (17.8%) | 232 (59.8%) | $p < 0.001$ *** |
| | Graduate school | 65 (16.8%) | 55 (14.2%) | 15 (3.9%) | 135 (34.8%) | |
| Monthly income (USD) | Below 1700 | 39 (10.1%) | 3 (0.8%) | 36 (9.3%) | 78 (20.2%) | |
| | Above 1700~below 2550 | 75 (19.4%) | 36 (9.3%) | 39 (10.1%) | 150 (38.9%) | $\chi^2$ = 75.001 |
| | Above 2550~blow 3400 | 35 (9.1%) | 33 (8.5%) | 8 (2.1% | 76 (19.7%) | $p < 0.001$ *** |
| | Above 3400~blow 4200 | 16 (4.1%) | 15 (3.9%) | 2 (0.5%) | 33 (8.5%) | |
| | Above 4200 | 18 (4.7%) | 28 (7.3%) | 3 (0.8%) | 49 (12.7%) | |
| Age | 20–29 | 46 (11.9%) | 8 (2.1%) | 39 (10.1%) | 93 (24.0%) | |
| | 30–39 | 84 (21.6%) | 48 (12.4%) | 42 (10.8%) | 174 (44.8%) | $\chi^2$ = 59.964 |
| | 40–49 | 41 (10.6%) | 43 (11.1%) | 6 (1.5%) | 90 (23.2%) | $p < 0.001$ *** |
| | more than 50 | 12 (3.1%) | 17 (4.4%) | 2 (0.5%) | 31 (8.0%) | |
| Total | | 100% | 100% | 100% | | |

** $p < 0.01$, *** $p < 0.001$.

## 5. Conclusions

As the convention industry has been recognized as a high value added business, the South Korean government and business have committed to enhancing infrastructure to have business events such as convention center construction, resort complex development, and convention district development, etc. [46] in the convention industry, yet the development of convention-related professionals and investment in human resources are relatively low [41]. The research examines the job characteristics and core competences of convention PM who needs more sophisticated professional skills and knowledges for HRM and competitiveness.

The first purpose of this study is to examine convention project managers' job characteristics through the preceding studies and supplement insufficient parts. Centering around the existing model of job characteristics [15], this study would draw the job characteristics appropriate for convention business, adding professionalism, a characteristic of convention business [22]. The job characteristics theory of Hackman & Oldham is an important theory recognized by researchers in the field of human resources management, but it is under criticism. Job characteristics of convention PM has been derived including the "professionalism" element. Based on the literature, convention project managers' job characteristics were drawn. Through factor analysis, five job characteristics of project managers in the convention field were identified, including professionalism/skill diversity, autonomy, task identity, task importance, and feedback. It shows that this study has buttressed the JCM empirically [47].

Secondly, this study is to draw the core competencies of convention project managers, which have not been considered. From the literature review, the initial dimensions were identified and the scale of core competencies was developed with in-depth interview. Crisis management ability, event attraction ability, task management ability, and communication ability were identified as convention project managers' core competencies, which support the preceding studies [48,49]. To enhance the job characteristics, the education and training for core competency is required. Specifically, it is necessary to enhance crisis management/communication skills like communication with other stakeholders including organizers and service providers. Also, it should be required that we promote competency-based training such as sales and negotiation, schedule arrangement, budget

planning, and management ability. However, as mentioned above, high core competency does not always lower turnover intention, and enhance job satisfaction and organizational commitment, it is necessary to proceed with it in the direction to lead the organization's effectiveness, improving core competencies by thoroughly understand the relationship of impact. A convention PM should have good sales and negotiation skills as a core competency to attract events. Also, the result supports the academic studies that marketing is one of the conference organizer's leadership qualities [50].

Lastly, this study scrutinizes the perception of core competency according to the job characteristics groups of the convention PM. Three groups were drawn, and the characteristics of the group with high perception of job characteristics were male, unmarried, college graduates, and those in their 30s. As for the characteristics of the group with low job characteristics, the proportion of males to females was similar; there were more married people; and their age and education levels were relatively high. To look at the perception of the importance of core competency, the group with low job characteristics presents that core competency is important. Low perception of job characteristics means that in the group satisfaction with the organization and the effectiveness of the organization is relatively low, and that motivation is required. This emphasizes the importance of core competency as appeared to the literature [51]. Specifically, convention project managers' turnover rate becomes higher as their careers as experts are accumulated. Also, the group of people with low perception of job characteristics have high in experience and career. For that reason, it is required that they have core competency training to enhance motivation and satisfaction as a career is accumulated in the organization.

This study was started for the purpose of the development of sustainable human resources with convention project managers. One of the limitations of this study comes from the Sampling issue. There is a limitation that this study could not consider the different level of company size and types of organization. The company's size and the project manager's level (Career and personal ability, etc.) may differ, so it would be beneficial to conduct a follow-up study reflecting this. Lastly, it is necessary to design and implement an actual education program, understanding the drawn core competencies and education needs. It would be necessary to continue to conduct research, reflecting this more elaborately in the future. Research should continue for convention project managers' sustainable human resources management. In addition, the convention business field has focused on quantitative development and the development of size, and in spite of being an industry with high dependence on manpower, there have been insufficient studies of human resources. It would be necessary to conduct education for the sake of improving core competency for each stage, analyzing career development path of employees in the convention industry, and continuously conducting research on measures for lowering turnover rate and increasing loyalty to the organization.

**Author Contributions:** Conceptualization, Y.Y., W.Y. and E.K.; methodology, Y.Y. and E.K.; software, W.Y.; validation, W.Y.; formal analysis, Y.Y. and E.K.; investigation, Y.Y. and E.K.; writing—original draft preparation, Y.Y., W.Y. and E.K.; writing—review and editing, Y.Y. and E.K.; supervision, Y.Y. and E.K. All authors have read and agreed to the published version of the manuscript.

**Funding:** This research received no external funding.

**Conflicts of Interest:** The authors declare no conflict of interest.

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
