# Peer review of "Towards Sustainable Human Resource Development of Convention Project Managers: Job Characteristics and Related Differences in Core Competency"

_sustainability, doi:10.3390/su12197898_

Round 1
Reviewer 1 Report
This submission is an empirical study that delves into a very important and interesting question in the MICE industry. The research question is logically derived from theory, and empirical analysis is done quite competitively.
The reviewer sees potential merit in the manuscript, but would like to raise the following issues in light of improving the paper.
1. It is necessary for the authors to highlight, in the Introduction, the point of differentiation and contribution of the study - what makes this paper different from other HR and MICE-related studies and how it constitutes merit.
2. Among the research questions in lines 271-278, Q1 and Q4 seem to be inconsistent with the context. These need to be corrected.
3. 3.2 Generation of Sample Items -> Generation of Questionnaire Items
Also, while the approach seems very robust, please provide additional details of the expert interview.
4. Please check again to make sure that reference style complies with the journal guidelines.
5. It is suggested that the title is rephrased as:
"Towards Sustainable Human Resource Development of Convention Project Managers: Job Characteristics and Related Differences in Core Competency"
Author Response
- It is necessary for the authors to highlight, in the Introduction, the point of differentiation and contribution of the study - what makes this paper different from other HR and MICE-related studies and how it constitutes merit.
Response: We thank the reviewers for their opinions, and added expressions by compressing the overall differentiation of the study. (lines 81-85) ]. The difference of this study is that it is an initial attempt to introduce the theory of job characteristics for convention project managers in the convention field where research on human resources is insufficient. In addition, in the field of human resource management and development, it is possible to derive academic implications that can be strengthened by verifying theories.
- Among the research questions in lines 271-278, Q1 and Q4 seem to be inconsistent with the context. These need to be corrected.
Response: We appreciated to the reviewers for his suggestion. These parts have now been modified (Lines 235-241).
The purpose of this study was to define the key variables for the job characteristics and core competencies of convention project managers, and to identify core competencies based on the type of job characteristics model of convention project managers. In order to do this, the study was designed to answer the following research questions:
Q1. Which factors are identified as the job characteristics and core competencies of convention project manager?
Q2. Are the core competences of convention projects manager different among groups of job characteristic?
Q3. Are the core competencies of convention project managers different among groups of job characteristics?
Q4. Are the demographics of convention project managers different among groups of job characteristics?
- 3.2 Generation of Sample Items -> Generation of Questionnaire Items
Also, while the approach seems very robust, please provide additional details of the expert interview.
Response: : Thank you for the comment. The text has been added at lines 259-262. A total of two interviews were conducted based on previous studies. In the first interview, a total of 20 convention project managers were confronted to collect opinions on core competencies. After coding and analyzing the data obtained through this, the core competencies were confirmed through 11 people in the second interview.
- Please check again to make sure that reference style complies with the journal guidelines.
Response: We have revised reference style according to the MDPI Author’s Guidelines.
- It is suggested that the title is rephrased as:
"Towards Sustainable Human Resource Development of Convention Project Managers: Job Characteristics and Related Differences in Core Competency"
Response: The title was revised to reflect the opinions of the reviewers. “Towards Sustainable Human Resource Development of Convention Project Managers: Job Characteristics and Related Differences in Core Competency”
Thanks for the accurate and keen review. Thanks to the reviewers, the quality of the paper has improved even further. Thank you very much.
Reviewer 2 Report
Dear Authors,
Please find my comments and suggestions below.
- The title should be shortened to be more suggestive.
- “Convention” used in the title and abstract shall be clearly defined from the beginning (in the abstract and in the introduction).
- The abstract shall end with very concise conclusions which shall give more information to the reader. In my view, it is too general.
- “the job characteristics theory” needs referencing – line 72
- Parts 2.1. and 2.2 shall contain figures/schemes/tables presenting clearly the job characteristics and competencies, respectively. Or at least present them in a more structure manner, with bullets and numbering, for example.
- Lines 184-185 “Generally, competency is divided into competency at 184 an individual level and competency at an organization level” need referencing or a better explanation. For me, it seems this phrase is an assumption of the authors.
- In my view, the whole chapter 2 shall be shortened to two thirds.
- The manuscript needs language check for mistakes and errors. Example: Q2 at line 273-274 is unclear.
- Line 287 – how many interviews did you conduct at this phase?
- “Measurement instruments” at lines 293-294 is unclear. Actually, I recommend restructuring the text from lines 280-301 to clearly present the methodology of research. The title of the section is unclear, too.
- Why did you perform all the statistical tests presented at lines 318-320? Please explain their importance for your research. Also, please explain in the paper if the results obtained (for example at lines 346-347) are significant and justify why. But avoid very general phrases as for example, lines 392-393.
- Please present, between brackets, all the sums of money in US dollars or Euros – much simpler for international readers to understand the paper – lines 327-328, 413 and 418.
- Please put tables and figures after the text, not before.
- For increasing visibility, please change the titles of sub-sections 4.2, 4.3. and 4.4 into more suggestive ones.
- Line 338 – the paper shall present how you generated the factors in Table 2. Actually, I did not understand the generation of the variables, as research instruments are not described in your paper.
- I could not find Table 7 – line 398
- The paper needs data on the economic value of convention industry describing its role within the South Korean economy – lines 422-423 seem to be assumptions. Such data shall form section 3.1 and then, start with research methodology.
- I consider that this study needs some referencing from more recent studies, if published in 2020.
- Please revise and respect MDPI Author’s Guidelines accordingly.
Yours faithfully,
Author Response
- The title should be shortened to be more suggestive.
Response: The title has been revised as to “Toward Sustainable HRM of Convention Project Manager: Differences in Core Competency based on the Types of Job Characteristics”
- “Convention” used in the title and abstract shall be clearly defined from the beginning (in the abstract and in the introduction).
Response: The text has been added (Lines 33-35).
- The abstract shall end with very concise conclusions which shall give more information to the reader. In my view, it is too general.
Response: The abstract has been revised and the sentence has been added in revised manuscript (Lines 20-23).
- “the job characteristics theory” needs referencing – line 72
Response: The reference has been added (Lines 72-73)
- Parts 2.1. and 2.2 shall contain figures/schemes/tables presenting clearly the job characteristics and competencies, respectively. Or at least present them in a more structure manner, with bullets and numbering, for example.
Response: The figure has been added in the revised manuscript as suggested (Lines 96 – 99).
- Lines 184-185 “Generally, competency is divided into competency at 184 an individual level and competency at an organization level” need referencing or a better explanation. For me, it seems this phrase is an assumption of the authors.
Response: We have appreciated the comment by the reviewer that we followed very carefully. As reviewer suggested the whole literature review chapter is refined and the sentence is deleted.
- In my view, the whole chapter 2 shall be shortened to two thirds.
Response: We agree with this suggestion. For this, we shortened the literature review part (page 2-5).
- The manuscript needs language check for mistakes and errors. Example: Q2 at line 273-274 is unclear.
Response: We appreciated to the reviewers for his suggestion. These parts have now been modified (Lines 240-241).
- Line 287 – how many interviews did you conduct at this phase?
Response: The text has been added at lines 259-262.
- “Measurement instruments” at lines 293-294 is unclear. Actually, I recommend restructuring the text from lines 280-301 to clearly present the methodology of research. The title of the section is unclear, too.
Response: Thank you for your comments. We have added measurement instruments at line 270-279.
- Why did you perform all the statistical tests presented at lines 318-320? Please explain their importance for your research. Also, please explain in the paper if the results obtained (for example at lines 346-347) are significant and justify why. But avoid very general phrases as for example, lines 392-393.
Response: The paragraph has been revised (page 6-7, Lines 297-306).
- Please present, between brackets, all the sums of money in US dollars or Euros – much simpler for international readers to understand the paper – lines 327-328, 413 and 418.
Response: The currency has been changed to US dollars. (Line 318, 412)
- Please put tables and figures after the text, not before.
Response: We have followed reviewer’s suggestion. the tables and figures are now been moved after text. (lines 307-413)
- For increasing visibility, please change the titles of sub-sections 4.2, 4.3. and 4.4 into more suggestive ones.
Response: The titles of sub-section has been revised (line 320, 362, 386, 396)
- Line 338 – the paper shall present how you generated the factors in Table 2. Actually, I did not understand the generation of the variables, as research instruments are not described in your paper.
Response: In response to the opinions of the reviewers, how the factors were derived was added to the front and revised so that the reader could understand them well.(lines 270-279)
- I could not find Table 7 – line 398
Response: Thank you. The table 8 is now renamed to table 7 (395)
- The paper needs data on the economic value of convention industry describing its role within the South Korean economy – lines 422-423 seem to be assumptions. Such data shall form section 3.1 and then, start with research methodology.
Response: By adding references according to the opinion of the reviewers, the persuasive power of the thesis was enhanced.(line 418)
- I consider that this study needs some referencing from more recent studies, if published in 2020.
Response: We agree with the reviewer’s suggestion that more recent reference needed. We have therefore added 3 new references below in the reference list (Reference #2, #23, #28).
- Lee, H.; Lee, J. An exploratory study of factors that exhibition organizers look for when selecting convention and exhibition centers. J. Travel Tour. Mark. 2017, 34, 1001–1017.
- Hwang, J. ; Jang, W. The effects of job characteristics on perceived organizational identification and job satisfaction of the Organizing Committee for the Olympic Games employees, Managing Sport and Leisure, 2020, 25(4), 290-306, DOI: 10.1080/23750472.2020.1723435
- Parker, S. K., Morgeson, F. P., ; Johns, G. One hundred years of work design research: Looking back and looking forward. Journal of applied psychology, 2017, 102(3), 403.
- Please revise and respect MDPI Author’s Guidelines accordingly.
Revised: Thank you. We have revised manuscript according to the MDPI Author’s Guidelines.
Thanks for the accurate and keen review. Thanks to the reviewers, the quality of the paper has improved even further. Thank you very much.
Round 2
Reviewer 2 Report
Dear Authors,
The only issue is the title, again. I hope that you will find a shorter and highly suggestive title.
Regards,